theoretical biology/applied mathematics/ biomathematics

mathematical modelling, DNA damage, adaptation, repair, heterogeneity, bet-hedging

**Authors for correspondence:**
Pierre Roux
e-mail: pierre.roux@universite-paris-saclay.fr
Delphine Salort
e-mail: delphine.salort@upmc.fr
Zhou Xu
e-mail: zhou.xu@sorbonne-universite.fr

# Adaptation to DNA damage as a bet-hedging mechanism in a fluctuating environment

Pierre Roux[1,2], Delphine Salort[1] and Zhou Xu[1]

[1]Sorbonne Université, CNRS, UMR7238, Institut de Biologie Paris-Seine, Laboratory of Computational and Quantitative Biology, 75005 Paris, France
[2]Laboratoire de Mathématiques d'Orsay (LMO), Université Paris-Sud, Paris-Saclay, Orsay, France

ZX, 0000-0001-9468-1406

In response to DNA damage, efficient repair is essential for cell survival and genome integrity. In eukaryotes, the DNA damage checkpoint is a signalling pathway that coordinates this response and arrests the cell cycle to provide time for repair. However, when repair fails or when the damage is not repairable, cells can eventually bypass the DNA damage checkpoint and undergo cell division despite persistent damage, a process called adaptation to DNA damage. Interestingly, adaptation occurs with a delayed timing compared with repair and shows a large variation in time, two properties that may provide a survival advantage at the population level without interfering with repair. Here, we explore this idea by mathematically modelling cell survival in response to DNA damage and focusing on adaptation parameters. We find that the delayed adaptation timing indeed maximizes survival, but its heterogeneity is beneficial only in a fluctuating damage-inducing environment. Finally, we show that adaptation does not only contribute to survival but also to genome instability and mutations, which might represent another criterion for its selection throughout evolution. Overall, we propose that adaptation can act as a bet-hedging mechanism for cell survival in response to DNA damage.

## 1. Introduction

Cells are often faced with damage to their DNA resulting from either external or endogenous sources. In response, eukaryotes are equipped with sophisticated mechanisms to deal with the damage, collectively termed the DNA damage response (DDR) [1]. Double-strand breaks (DSBs) are particularly cytotoxic and have been used as a model to dissect the DDR, notably in the yeast *Saccharomomyces cerevisiae*. Following a DSB, the early steps of the DDR lead to the recruitment of the protein kinases Tel1 and Mec1 to the damage, which initiate the DNA damage checkpoint signalling pathway. A major function of the

checkpoint, executed by the effector kinases Rad53 and Chk1, is to arrest the cell cycle at the G2/M phase, preventing cell division while the damage is present. The DDR also induces the activation of appropriate repair pathways, mainly non-homologous end-joining (NHEJ) and homologous recombination (HR), depending on the phase of the cell cycle and the existence of sequences homologous to the break site. In specific contexts, other repair mechanisms can be used by the cell. The different repair pathways have different kinetics, with break-induced replication (BIR), a mechanism that heals one-ended breaks by copying a template chromosome arm to its extremity, possibly being one of the slowest (typically approx. 5 h) [2]. BIR is also much less frequently used by the cell and most of the repair for a DSB is therefore done within 5 h after damage. If the DSB is correctly repaired, cells recover normal cell cycle progression with an intact genome. Defective repair can lead to mutations and genome rearrangement in the progeny.

However, appropriate repair pathways might not always be available and the DDR might, instead of repair, steer the cell towards other cell fates such as cell death or adaptation. Here, adaptation to DNA damage is defined as the process of allowing cell cycle progression and cell division despite the initial checkpoint activation and persistent damage [3–5]. During adaptation, the effector branch of the checkpoint is inactivated, releasing the cell from arrest. While the detailed molecular mechanisms of adaptation are not completely sorted out, it is established that adaptation is an active mechanism that can be genetically manipulated and can occur in most unrepaired cells in certain experimental settings, with a very variable timing ranging from 5 to 15 h [6].

It is speculated that adaptation might be a last resort mechanism for cell survival when all repair options or attempts are exhausted. Adaptation might then allow one of the two daughter cells to survive in case of asymmetric segregation of the damage or by activating alternative repair pathways available at another phase of the cell cycle [7,8]. Consistently, the survival of irradiated HR-deficient cells, which are unable to repair, is decreased in a strain with an adaptation-deficient allele of the Polo kinase *CDC5*, *cdc5-ad* [9]. However, cell division with an unrepaired DSB leads to improper chromosome segregation [10]. More generally, adaptation is associated with elevated chromosomal instability [9,11]. Interestingly, in an experiment where telomere deprotection, which is detected by the cell as an unrepairable damage, can be controlled by temperature using a thermosensitive mutant (*cdc13-1*), returning the cells to permissive temperature after a period of time at restrictive temperature is associated with better cell survival when adaptation is prevented [12]. This result suggests that commitment to adaptation entails a high mortality rate associated with chromosome missegregation. In other words, in a favourable environment when repair is possible, cells that do not adapt or have not yet adapted survive better.

The separation of timing between repair and adaptation naturally creates a hierarchy of cell fate decisions, repair being attempted first and leading to better survival and less genome instability. Because the exhaustion of repair options is not a single deterministic event, it is also attractive to speculate that the timing heterogeneity of adaptation might represent a bet-hedging mechanism to maximize overall survival. Bet-hedging can be defined as the maximization of long-term fitness in an unpredictable environment at the cost of a suboptimal fitness in any given environment, for an isogenic population [13,14]. This can be achieved either by a single suboptimal but not completely maladaptive phenotype or by a diversity of phenotypes that spreads the risk [13]. The unpredictable fluctuation of the environment, which can prevent or on the contrary facilitate repair, is a crucial factor to take into account in this bet-hedging hypothesis for adaptation. For example, cells that are subjected to constant DNA damage from X-ray irradiation might not efficiently repair if the repair capacity is overloaded. If the source of damage persists for a long time, adaptation might represent a chance for the population to survive despite a high mortality rate. However, if irradiation suddenly stops, cell survival at the population level might be increased if some cells have not yet committed to adaptation because they would then be able to repair. A heterogeneous adaptation timing might therefore be an optimal cell survival strategy.

In this work, we investigate the consequences of adaptation timing and heterogeneity in the survival of a population of isogenic yeast cells subjected to DNA damage, for which adaptation is best characterized. We ask if known properties of adaptation, namely slow kinetics and variable timing, represent a selective advantage and in which contexts. To do so, we use a stochastic compartment model of damaged cells evolving over time. We then develop a continuous version of the model based on ordinary differential equations and reproducing the overall properties of the stochastic model. Our modelling and numerical simulations suggest that correct adaptation timing is important for maximal cell survival and that adaptation heterogeneity might be explained as a bet-hedging survival strategy in fluctuating environments.

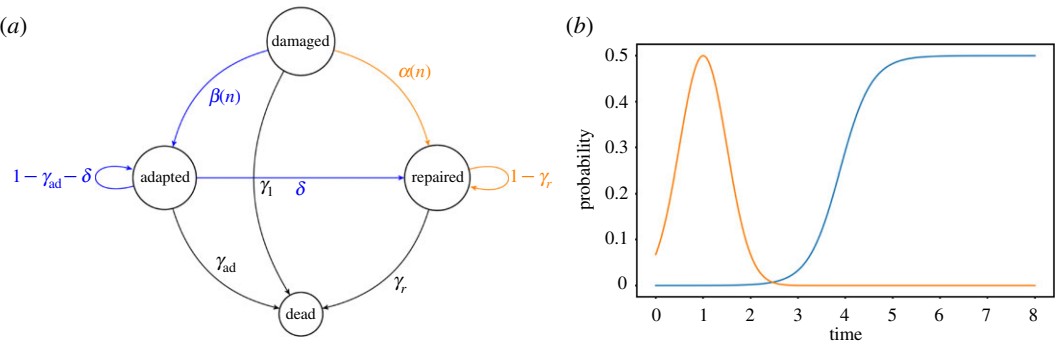

**Figure 1.** (*a*) Diagram of the Markovian process controlling the cellular states. (*b*) Probabilities of adaptation (blue) and repair (orange) depending on the time elapsed since the initial damage occurred, with $\alpha_m = 0.5$, $\sigma = 0.5$, $\mu_a = 1$ and $\beta_m = 0.5$, $p = 3$, $\mu_b = 3.886$.

## 2. Results

### 2.1. Mathematical model

#### 2.1.1. Stochastic model

To investigate the survival of a population of damaged cells, we first build a stochastic model that mimics the changes of individual cell states over discrete time. A step of this discrete time is chosen to be equivalent to the duration of a normal yeast cell cycle in optimal growth conditions (approx. 1.5–2 h). In this model, we consider an initial population of $N_0$ cells in which DNA damage is induced only once at time 0 and which activates the DNA damage checkpoint. The cells are therefore arrested and face three main cell fates: recovery after repair, adaptation or cell death (figure 1*a*). In turn, adapted cells can continue to adapt, undergo repair or cell death. The fact that adapted cells can repair a damage that was not repaired in the previous cell cycle reflects the idea that alternative repair pathways might be activated at other phases of the cell cycle [7,8]. We fix repair and cell death probabilities and ask how adaptation can impact overall cell survival. We then monitor the number of repaired cells and their descendants, collectively called healthy cells, since they are essential for long-term survival of the population. The main output measurement of our simulations is the time it takes for the environment to be entirely filled of healthy cells, i.e. the number of steps it takes to have $N_{\max}$ healthy cells in the population, which we call saturation time $T_S$. The maximum number of healthy cells constraint $N_{\max} > N_0$ is chosen both for computational purpose and to focus on the survival pattern at early times and not on long-term growth. Throughout the work, plots related to repair are colour-coded in orange and those related to adaptation in blue.

We take into account the following mechanisms (figure 1*a*):

— at each discrete step, damaged cells have a probability $\gamma_1$ of dying;
— depending on the number $n$ of steps since the damage, damaged cells adapt with a probability $\beta(n)$ and repair their DNA with a probability $\alpha(n)$;
— at each step, adapted cells die with probability $\gamma_{\mathrm{ad}}$ and repair their DNA with probability $\delta$;
— at each step, repaired cells die with probability $\gamma_r$;
— at the end of each step, the adapted and repaired cells divide within the limit of the carrying capacity of the environment. If there is not enough space to double both adapted and repaired populations, we fill the environment proportionally.

Mathematically speaking, the aforementioned process is Markovian for a suitable choice of (infinite) states space. Moreover, each individual cell evolves through an infinite states space Markov chain, or more simply through a finite states space time-varying Markov chain.

We assume the probability of repairing at the $n$th step to be given by a Gaussian centred on $\mu_a > 0$ and of variance $\sigma > 0$ (figure 1*b*),

$$\alpha(n) = \alpha_m \, \mathbf{e}^{-(n-\mu_a)^2/\sigma^2}.$$

The parameter $\alpha_m$ represents the maximum value of the probability of repair.

Adaptation probability at the $n$th step after damage is represented by a logistic curve, consistent with experimental data [6], parametrized the following way (figure 1$b$):

$$\beta(n) = \frac{\beta_m}{1 + e^{-p(n - \mu_b)}},$$

with coefficients $\mu_b$, $p$, $\beta_m$. The parameter $p$ represents the slope of the curve: the larger $p$ is, the faster adaptation goes from a low to a high probability with increasing steps. The parameter $\beta_m$ represents the maximum value of the probability of adaptation. Lastly, the central value $\mu_b$ of the logistic curve represents the moment cells adapt with half the maximum probability.

Note that $\alpha$ and $\beta$ are not probability densities, but probabilities computed at every step with the given formula; in particular, their sum over all integers $n > 0$ is not 1.

In order to perform numerical simulations of this model, we fix some of the parameters:

— population parameters: $N_0 = 20\,000$, $N_{max} = 100\,000$;
— death probability parameters: $\gamma_1 = 0.1$, $\gamma_{ad} = 0.35$, $\gamma_r = 0$;
— transition adapted → repaired: $\delta = 0.02$;
— parameters for $\alpha$: $\alpha_m = 0.5$, $\sigma = 0.5$, $\mu_a = 1$;
— parameters for $\beta$: $\beta_m = 0.5$, $p = 3$.

The significant death rate of damaged cells will naturally tend to favour faster cellular responses to maximize survival. The high death rate for adapted cells reflects the risk for a cell to segregate damaged chromosomes. The death rate of repaired cells is considered negligible.

This stochastic model provides a direct representation of the biological events occurring in the population of cells over time, which is easy to interpret. For instance, it allows us to find the value of $\mu_b$ that maximizes survival (i.e. minimizes $T_S$) and compare it with the known timing of adaptation (see §2.2 and figures therein). However, to reach conclusive results, the large number of simulations required for robust averages are computationally intensive. We therefore develop a continuous model representing the natural derivation and recapitulating the average behaviour of the stochastic model.

### 2.1.2. Continuous model

The continuous model is built upon a set of ordinary differential equations, which allows for fast numerical simulation of the average behaviour of the system.

For a given time $t > 0$, we denote $D(t)$ the quantity of damaged cells, $A(t)$ the quantity of adapted cells and $R(t)$ the quantity of repaired cells.

In order to build a continuous model, we use the natural interpolation of the stochastic model.

— The reproduction part is replaced with logistic growth terms

$$A(t)\left(1 - \frac{A(t) + R(t) + D(t)}{N_{max}}\right)$$

and

$$R(t)\left(1 - \frac{A(t) + R(t) + D(t)}{N_{max}}\right).$$

— In the continuous time between two steps, we use the natural continuous interpolations for $\alpha(t)$ and $\beta(t)$,

$$\alpha(t) = \alpha_m\, e^{-(t - \mu_a)^2 / \sigma^2}$$

and

$$\beta(t) = \frac{\beta_m}{1 + e^{-p(t - \mu_b)}}.$$

— The continuous transitions are written according to the classic equivalence between finite states space Markov chains and linear differential equations.

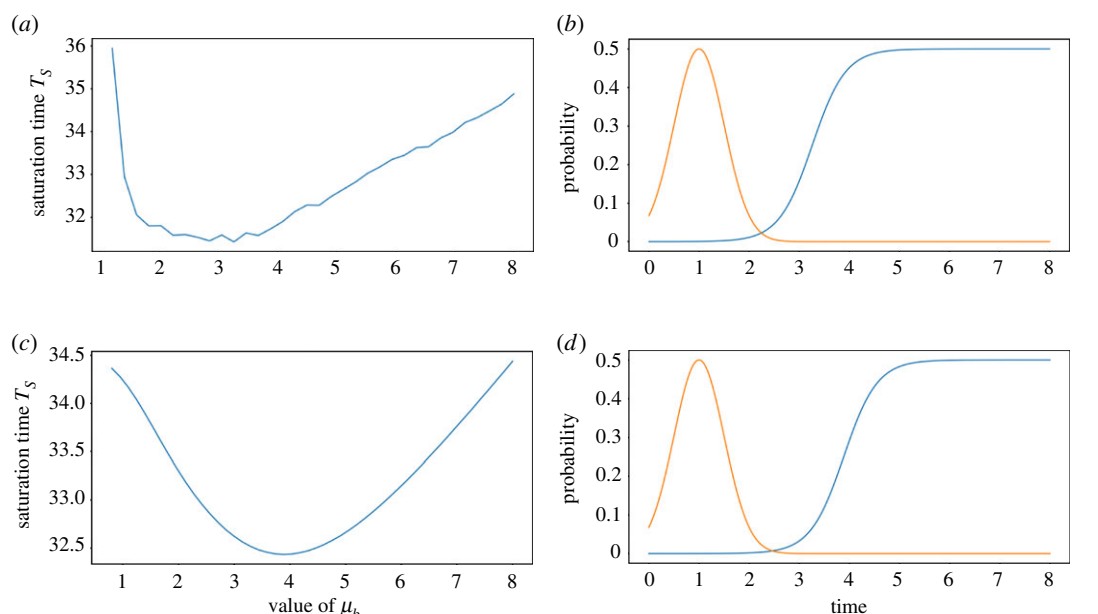

**Figure 2.** (a) Mean saturation time $T_S$ as a function of $\mu_b$ in the stochastic model, for $p = 3$. The means are taken over 10 000 simulations for each point. (b) Adaptation probability for the optimal $\mu_b$ (blue) found in (a) and fixed repair probability (orange). (c) Saturation time $T_S$ as a function of the value of $\mu_b$ for the deterministic model, for $p = 3$. (d) Adaptation probability for the optimal $\mu_b$ (blue) found in (c) and fixed repair probability (orange).

In short, the model writes

$$
\frac{\mathrm{d}}{\mathrm{d}t}
\begin{pmatrix} D(t) \\ A(t) \\ R(t) \end{pmatrix}
=
\begin{pmatrix}
-\gamma_1 - \beta(t) - \alpha(t) & 0 & 0 \\
\beta(t) & -\gamma_{\mathrm{ad}} - \delta & 0 \\
\alpha(t) & \delta & -\gamma_r
\end{pmatrix}
\begin{pmatrix} D(t) \\ A(t) \\ R(t) \end{pmatrix}
+
\begin{pmatrix}
0 \\
A(t)(1 - \frac{A(t)+R(t)+D(t)}{N_{\max}}) \\
R(t)(1 - \frac{A(t)+R(t)+D(t)}{N_{\max}})
\end{pmatrix}.
$$

Unless otherwise specified, we use the following values in the numerical experiments, which are the same as for the stochastic model:

— population parameters: $N_0 = 20\,000$, $N_{\max} = 100\,000$;
— death probability parameters: $\gamma_1 = 0.1$, $\gamma_{\mathrm{ad}} = 0.35$, $\gamma_r = 0$;
— transition adapted $\rightarrow$ repaired: $\delta = 0.02$;
— parameters for $\alpha$: $\alpha_m = 0.5$, $\sigma = 0.5$, $\mu_a = 1$;
— parameters for $\beta$: $\beta_m = 0.5$, $p = 3$.

Note that with these values, the slope of the curve $\beta$ at its centre $\mu_b$ is

$$
\beta'(\mu_b) = \frac{p\beta_m}{4} = \frac{p}{8}.
$$

For this reason, we say by abuse of notation that $p$ describes the slope of the adaptation curve.

## 2.2. Optimal adaptation timing in response to repairable DNA damage

We first ask whether tuning adaptation timing may maximize cell survival, all other parameters being equal. To do so, we test several values of $\mu_b$ in the stochastic model and for each of these we measure the average of the saturation times $T_S$ it takes for the population to be filled with $N_{\max}$ healthy cells. These mean times for the different values of $\mu_b$ are computed by simulating the model 10 000 times (figure 2a). We find that a minimum for $T_S$ is reached for a value of $\mu_b$ close to the tail of the repair probability (figure 2b).

We then perform a similar simulation using the continuous model. For a given parameter $\mu_b$, we now define the saturation time $T_S$ as the time it takes for the population to satisfy

$$
|N_{\max} - R(t)| < 1.
$$

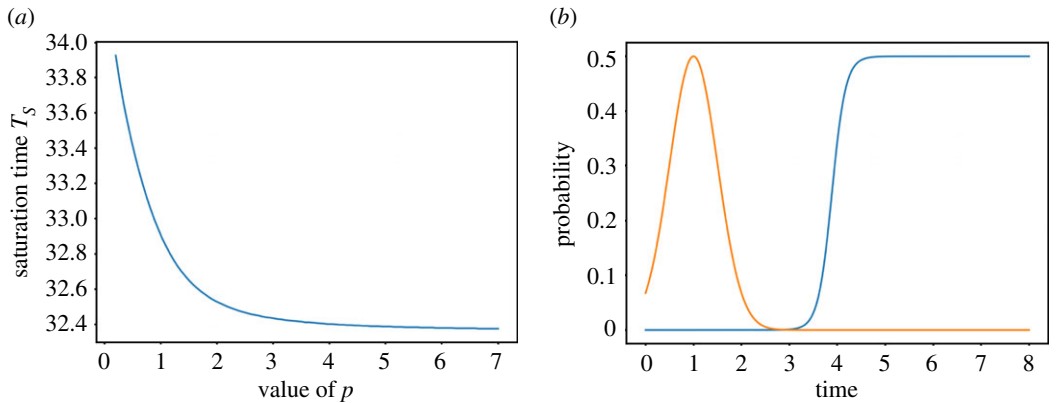

**Figure 3.** (*a*) Saturation time $T_S$ as a function of $p$ for $\mu_b = 3.886$. (*b*) Adaptation probability for the optimal $p$ (blue) and fixed repair probability (orange).

This criterion means that the environment is saturated at 99.9999% with healthy cells. We obtain a similar result, with a clear optimal value (figure 2*c*,*d*).

The values obtained with the stochastic and continuous models do not match exactly, because the latter allows the population to repair and adapt between discrete steps and thus the population grows slightly faster. Nevertheless, the deterministic model qualitatively recapitulates the stochastic model, with the major advantage of being computationally efficient. We therefore use the continuous model for the rest of this work.

The optimal $\mu_b$ found by simulating both models suggests that adapting as early as possible after repair leads to better survival. To test this idea, we repeat the simulation with different repair probabilities by varying $\mu_a$ and plotting the optimal $\mu_b$ as a function of $\mu_a$ (figure 6). We find that the optimal adaptation timing $\mu_b$ is delayed from $\mu_a$ so that cells adapt only after exhaustion of repair possibilities, thus setting a hierarchy of cell fate decisions.

## 2.3. Adaptation heterogeneity does not improve cell survival after a single initial damaging event

We next investigate the contribution of adaptation heterogeneity, represented by the slope $p$ of the adaptation probability, to cell survival. We set $\mu_b$ to the previously obtained optimal value and compute the saturation time $T_S$ as a function of $p$ (figure 3).

For this choice of $\mu_b$, we find that the largest tested value of $p$ maximizes cell survival. By contrast, a heterogeneous adaptation response closer to the experimentally measured one [6] (e.g. $p \approx 1$–3) gives a slower population growth after DNA damage (figure 3*a*).

To explore more exhaustively the effect of adaptation timing and heterogeneity, we plot $T_S$ in the parameter space ($\mu_b$, $p$) (figures 4, 7 and 8). We find that for $1.5 \leq \mu_b \leq 6$, which includes the previously obtained value, the optimal choice is to maximize $p$. For $\mu_b < 1.5$ and $\mu_b > 6$, a minimum can be found for $p \approx 0.5$ (figure 8). These two cases correspond to adaptation overlapping with repair and very late adaptation, respectively, where heterogeneity slightly compensates for suboptimal $\mu_b$. Indeed, the saturation time is much larger for these values of $\mu_b$ than for more realistic ones.

Except for these extreme cases ($\mu_b < 1.5$ and $\mu_b > 6$), which do not agree with experimental observations, our model indicates that adaptation heterogeneity is a suboptimal survival strategy in a context where damage is induced once at time point 0. We reason that because our model considers that no further damage is induced in the cells after the initial one, the environment becomes predictable and amenable to optimal survival strategy with no heterogeneity. By contrast, in a more realistic context, the initial source of damage might be active for an unpredictable amount of time. In the next section, we explore the idea that a fluctuating environment might favour a heterogeneous adaptation response.

## 2.4. The preference for sharp adaptation timing is erased in a fluctuating environment

To represent a fluctuating environment, the initial damage source continues to be present in the environment for an unpredictable period of time. We implement this scenario in three different manners (figures 5 and 9):

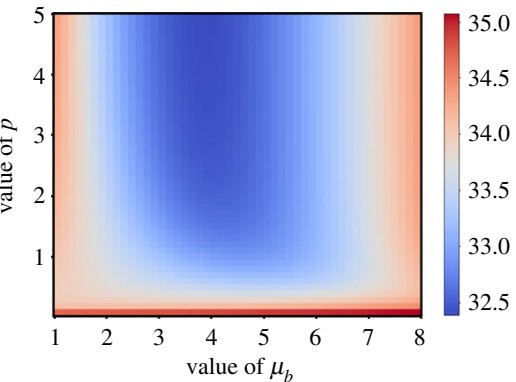

**Figure 4.** Saturation time $T_S$ displayed in colour gradient as a function of both $\mu_b$ and $p$.

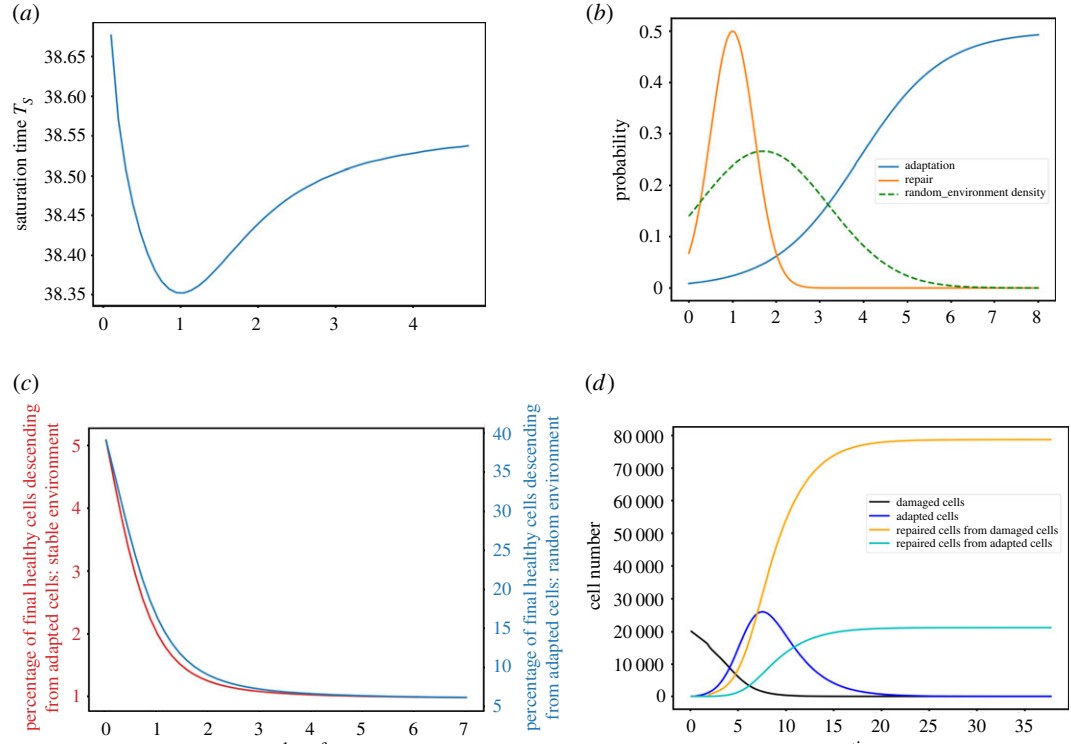

**Figure 5.** (*a*) Saturation time $T_S$ as a function of $p$ in a Gaussian random environment for the moment when the damage source stops with mean 1.7 and variance 2.25 (see (*b*)), for $\mu_b = 3.886$. (*b*) Adaptation probability for the optimal $p$ (blue) and fixed repair probability (orange). The dashed curve represents the Gaussian probability density of the random environment. (*c*) Fraction of cells (in %) in the final healthy population that are descended from adapted cells as a function of $p$, in a random (blue, right $y$-axis) or stable (red, left $y$-axis) environment. (*d*) Evolution in time of the four cell subpopulations (damaged cells, adapted cells, repaired cells directly from damaged cells and repaired cells descended from adapted cells), with $\mu_b = 3.886$ and $p = 1$, averaged over the distribution of the random environment.

— The moment when the source of damage stops follows a normal law; no repair is possible while the damage process is ongoing; repair resumes once the source of damage stops but with the initially fixed probability (figure 5*b*).
— Same as above, but with an exponential law (figure 9*b*).
— The moment when the source of damage stops follows a uniform law; no repair is possible while the damage process is ongoing; repair probability is initiated only once the source of damage stops and its maximum probability fades exponentially over time (figure 9*d*).

For each implementation of the fluctuating environment, we set $\mu_b = 3.886$ and compute $T_S$ as a function of $p$.

Strikingly, simulations in all three implementations lead to similar results with $T_S$ minimized for an optimal $p \approx 1$, which corresponds to a gentle slope in agreement with experimental observations (figures 5a,b and 9). However, closer examination indicates that $T_S$ varies much less as a function of $p$ than in the context of a stable environment and, therefore, its minimum represents only a minor improvement in cell survival compared with other values of $p$. We conclude that a fluctuating environment erases the preference for a sharp adaptation timing and only slightly select for adaptation heterogeneity. It is thus not clear whether maximizing cell survival alone might be sufficient to explain adaptation heterogeneity. In the next section, we ask whether another property of adaptation—that it induces genome instability—might be promoted by heterogeneity.

## 2.5. Adaptation heterogeneity promotes genetic diversity in response to DNA damage

The saturation time criterion we use in this work aims at measuring the moment when the population becomes large and healthy enough to be maintained. Therefore, any cell that has eventually repaired the initial damage is counted, whether it comes from the damaged cells pool or the adapted cells pool. Given that in a fluctuating environment, adaptation heterogeneity leads to only a minor improvement in fitness, we wonder whether different values of $p$ lead to different population structures, i.e. different fractions of healthy cells coming from the adapted pool. Since adaptation induces mutations and genome rearrangements, the fraction of the population of cells descending from adapted cells can be viewed as a proxy for the genetic diversity of the surviving population.

In order to have access to the structure of the final population, we divide the repaired compartment of the model in two parts:

$$R(t) = R_d(t) + R_a(t),$$

where $R_d(t)$ describes the amount of cells that directly repaired their DNA or are descended from cells which did and $R_a(t)$ describes the amount of cells that first underwent adaptation and then repair, or are descendant of these cells.

We denote

$$N(t) = A(t) + R_d(t) + R_a(t) + D(t),$$

the total population at time $t$. The extended model writes

$$\frac{d}{dt}\begin{pmatrix} D(t) \\ A(t) \\ R_d(t) \\ R_a(t) \end{pmatrix} = \begin{pmatrix} -\gamma_1 - \beta(t) - \alpha(t) & 0 & 0 & 0 \\ \beta(t) & -\gamma_{ad} - \delta & 0 & 0 \\ \alpha(t) & 0 & -\gamma_r & 0 \\ 0 & \delta & 0 & -\gamma_r \end{pmatrix} \begin{pmatrix} D(t) \\ A(t) \\ R_d(t) \\ R_a(t) \end{pmatrix} + \begin{pmatrix} 0 \\ A(t)\left(1 - \frac{N(t)}{N_{max}}\right) \\ R_d(t)\left(1 - \frac{N(t)}{N_{max}}\right) \\ R_a(t)\left(1 - \frac{N(t)}{N_{max}}\right) \end{pmatrix}.$$

We then compute for each value of the parameter $p$ the saturation time $T_S$ as above, and we estimate the ratio

$$\frac{R_a(T_S)}{R_d(T_S) + R_a(T_S)}.$$

The final fraction of healthy cells descended from adapted cells decreases as a function of $p$ and already reaches a plateau for $p > 3$ (figure 5c). In a fluctuating environment, this fraction is 17% for $p = 1$ (figure 5c, blue line). Thus, adaptation heterogeneity allows more progeny of adapted cells to be included in the surviving population, which should translate into a more genetically diverse population. For $p = 1$, we compute the evolution over time of all four subpopulations $A(t)$, $R_d(t)$, $R_a(t)$, $D(t)$ averaged over the distribution of the random environment (figure 5d). We find that adapted cells behave as a transient subpopulation that coexists with cells repaired directly from damaged cells, before being partially repaired themselves. A similar pattern is observed without fluctuating environment (figure 10a, for $p = 1$), but then the fraction of repaired cells descended from the adapted pool is much smaller (figure 5c). By contrast, in a context where no direct repair of damaged cells is available, adaptation followed by alternative repair pathways becomes the sole survival route for the population, which then recovers very slowly (figure 10b).

Overall, in a fluctuating environment, the results on population survival and structure suggest that evolution might have selected adaptation heterogeneity both to maximize survival and to promote genetic diversity.

# 3. Discussion

## 3.1. Optimal adaptation timing leads to a hierarchy of cell fate decisions

The DNA damage response is important to maintain cell survival and genome integrity. A plethora of repair mechanisms tailored to different types of damage act to fulfil this function. Interestingly, adaptation to DNA damage is used only after most or all repair attempts are exhausted. How can we explain this chronological priority of repair over adaptation? One possibility would be that adaptation is a passive response. The checkpoint could have been selected throughout evolution to robustly arrest the cell cycle only long enough for repair, after which its signal fades and allows for adaptation. However, in several adaptation mutants, including *cdc5-ad*, the checkpoint remains active for as long as 24 h [4,5,15,16], indicating that the checkpoint is intrinsically capable of maintaining cell cycle arrest for much longer than 5 h. Besides, the fact that overexpression of Cdc5 accelerates adaptation [6,17] suggests that it is an active mechanism. Another attractive possibility would posit adaptation as a process selected by evolution for its impact on cell survival after DNA damage in unicellular eukaryotes such as *S. cerevisiae*. Our work explores this scenario by mathematical modelling and numerical simulations. The conclusions might be applied to other eukaryotic microorganisms in which adaptation to DNA damage is conserved.

In a model with fixed repair probabilities, we ask whether a particular adaptation timing maximizes cell survival after DNA damage. We find a timing that agrees very well with experimental observations, corresponding to adaptation occurring just after most repair options are exhausted. This order of events is forced by the higher mortality rate of adapted cells compared with repaired cells. The high mortality rate of adapted cells is consistent with the missegregation of damaged chromosomes and possibly other yet undescribed mitotic abnormalities during adaptation [9,10]. We therefore suggest that maximizing cell survival is sufficient to create a hierarchy of cell fate decisions. Consistently, we found that even if repair timing is artificially changed, the optimal adaptation timing is shifted accordingly so that the delay between repair and adaptation is maintained.

How adaptation timing is implemented at the molecular level is not yet known. It would be interesting to decipher whether the average adaptation timing is set at an absolute value independently of other processes or whether it is actively regulated, for instance, by the DDR. Supporting the latter possibility, a number of factors involved in adaptation also regulate other aspects of the DDR, and for instance, the recombination protein Tid1 (also known as Rdh54) is a phosphorylation target of the DNA damage checkpoint and is important both for a subset of BIR that does not rely on the recombinase Rad51 and for adaptation [18–20]. Furthermore, repair in *tid*1Δ cells appears to be more efficient than in wild-type cells [2]. Thus, Tid1 might act at the crossroad of repair and adaptation, and regulate their respective timings. Another essential adaptation factor, Cdc5, potentially interacts with or phosphorylates a number of proteins involved in the DDR and in repair, including Mus81-Mms4, Sae2, Rad53 and Exo1 [6,17,21–23]. Although we found that the adaptation mutant *cdc5-ad* has no defect in homology-dependent repair [11], it was reported to be involved in the *RAD52*-independent repair of a dicentric chromosome [24], and it is thus attractive to speculate that Cdc5's function in adaptation might be affected by its potential involvement in the DDR.

## 3.2. Heterogeneity of adaptation timing: a putative bet-hedging strategy

While it is often straightforward to understand that the order of events can affect a biological function, it can be much more complicated to provide a functional basis for the heterogeneity of a biological process. Mathematical modelling is a powerful approach to test how heterogeneity affects a particular trait by artificially modulating heterogeneity, which is often difficult to achieve experimentally.

Here, we show that, in the defined context of a predictable environment with no prolonged source of damage, adaptation heterogeneity does not improve cell survival at the population level compared with a sharp adaptation timing. However, the 'repairability' of a damage is not always predictable. Indeed, activation of specific repair mechanisms depends upon many parameters, some of which possibly being stochastic or at least variable. The source of damage might also overload the repair machinery of the cell for an unpredictable amount of time. At the same time, adaptation is not the preferred outcome because of its associated mortality and genome instability, at least if repair is still possible. Therefore, to account for such an unpredictability, a risk-spreading strategy might increase long-term survival by increasing the timing heterogeneity of adaptation. We test this idea by

implementing different types of randomly changing environments that prolong the initial damaging phase. In each case, as predicted and in contrast to the context of a predictable environment, we find that a heterogeneous adaptation response maximizes cell survival. Closer examination of the saturation time values indicates that rather than strongly favouring heterogeneity, an unpredictable fluctuating environment actually erases the previously found preference for a sharp adaptation timing.

The phenotypic heterogeneity of adaptation timing can be the result of a bet-hedging strategy [25] as we propose or due to other modes of response, such as a sense-and-response strategy [26] whereby the damaged cell would monitor and evaluate its own repair ability and undergo adaptation if repair is deemed unfeasible or unsuccessful. While the examples of Tid1 and Cdc5 show that cross-talks might exist between repair and adaptation, the sense-and-response scenario to explain heterogeneity has been rejected by experiments in yeast where the cell population has no functional repair pathway available to begin with and still follows a delayed and heterogeneous adaptation timing [4,5,9]. Adaptation heterogeneity is therefore a built-in property, aimed at generating diversity independently of the repair status or the environment, rather than the result of a responsive strategy.

Overall, the heterogeneity of adaptation to DNA damage fits several criteria defining bet-hedging [13,14]: it is an intrinsic genetically encoded property observed at the level of the isogenic population and it is suboptimal in a constant environment but maximizes survival in unpredictable damaging environments. Further validation of this hypothesis might involve testing experimentally if yeast strains with different levels of adaptation heterogeneity (e.g. natural variants or engineered mutants) would show altered fitness in constant versus fluctuating environments.

In addition to a slight advantage for cell survival provided by a heterogeneous adaptation timing in a fluctuating environment, our results suggest that other factors might contribute to the selection of adaptation heterogeneity. Interestingly, we find that, for roughly equally fit populations, higher levels of adaptation heterogeneity lead to a greater fraction of the surviving population descended from adapted cells and having repaired their damage afterwards. We thus propose that adaptation heterogeneity allows adapted cells to contribute more to the genetic make-up of the surviving population in the wild. Starting from isogenic cells, adaptation-induced mutations and genome rearrangements generate genetic diversity. In conclusion, our model suggests that adaptation heterogeneity might have been selected both as a survival strategy and also as a driver of genetic diversity in populations subjected to unpredictable DNA damage.

Data accessibility. All codes used in this work are available at: https://github.com/pierreabelroux/Roux_Salort_Xu_2021/blob/main/code_RSX.py.

Authors' contributions. P.R. and D.S. built the mathematical models, with inputs from Z.X. P.R. performed the computations and simulations. P.R. and Z.X. drafted the manuscript. D.S. and Z.X. supervised the work. All authors revised the manuscript.

Competing interests. We declare we have no competing interests.

Funding. This work was supported by Ville de Paris (Programme Émergence(s)).

# Appendix A

See figures 6–10.

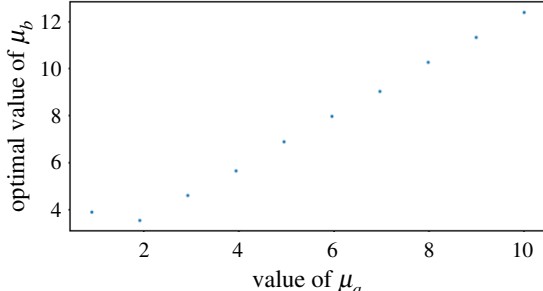

**Figure 6.** Optimal value for $\mu_b$ as a function of $\mu_a$, for $p = 3$.

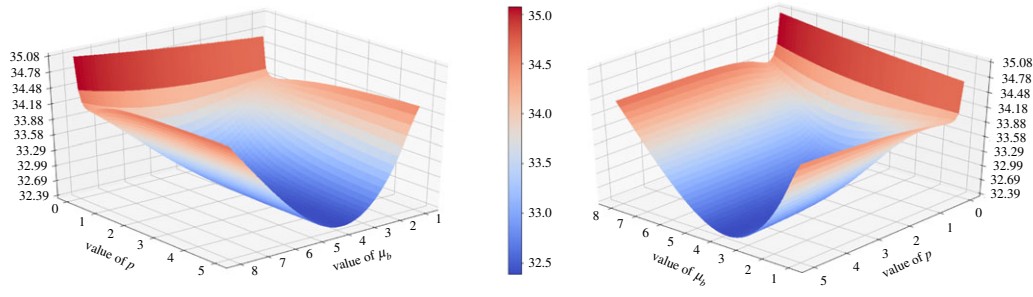

**Figure 7.** Saturation time $T_S$ displayed in colour gradient as a function of both $\mu_b$ and $p$. Two surface plot views are shown.

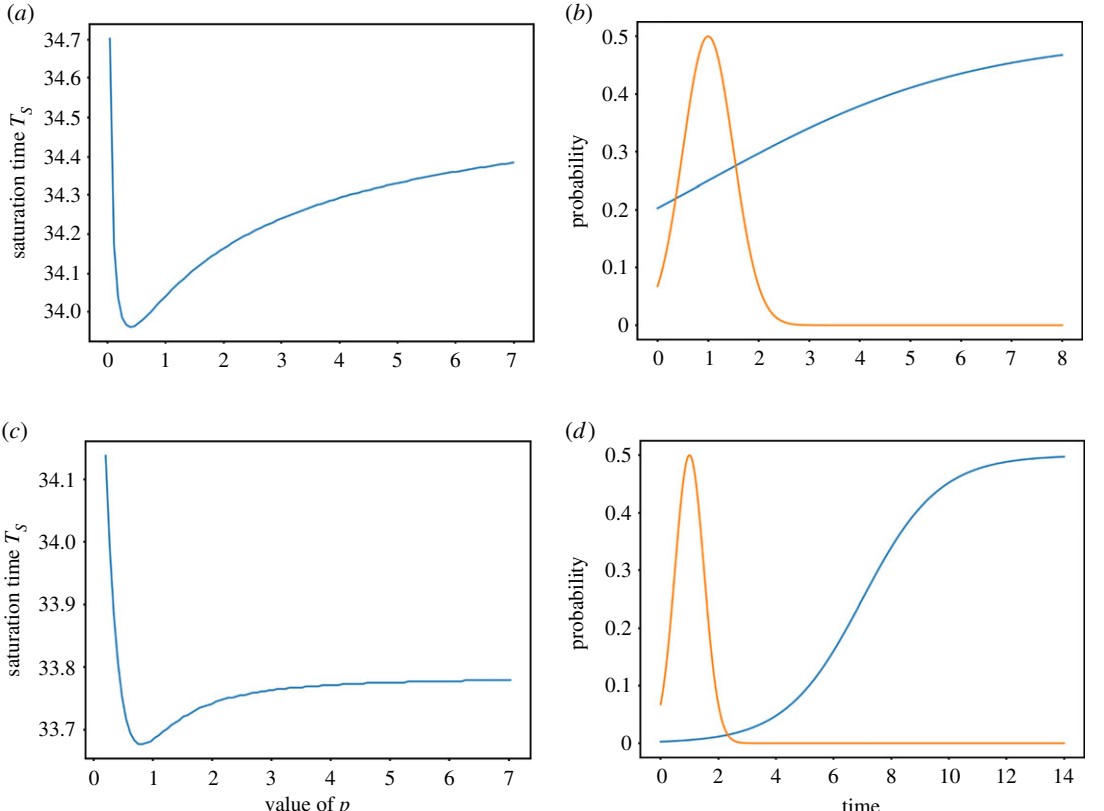

**Figure 8.** (a) Saturation time $T_S$ as a function of $p$, for $\mu_b = 1$. (b) Adaptation probability for the optimal $p$ from (a) (blue) and fixed repair probability (orange). (c) Saturation time $T_S$ as a function of $p$, for $\mu_b = 7$. (d) Adaptation probability for the optimal $p$ from (c) (blue) and fixed repair probability (orange).

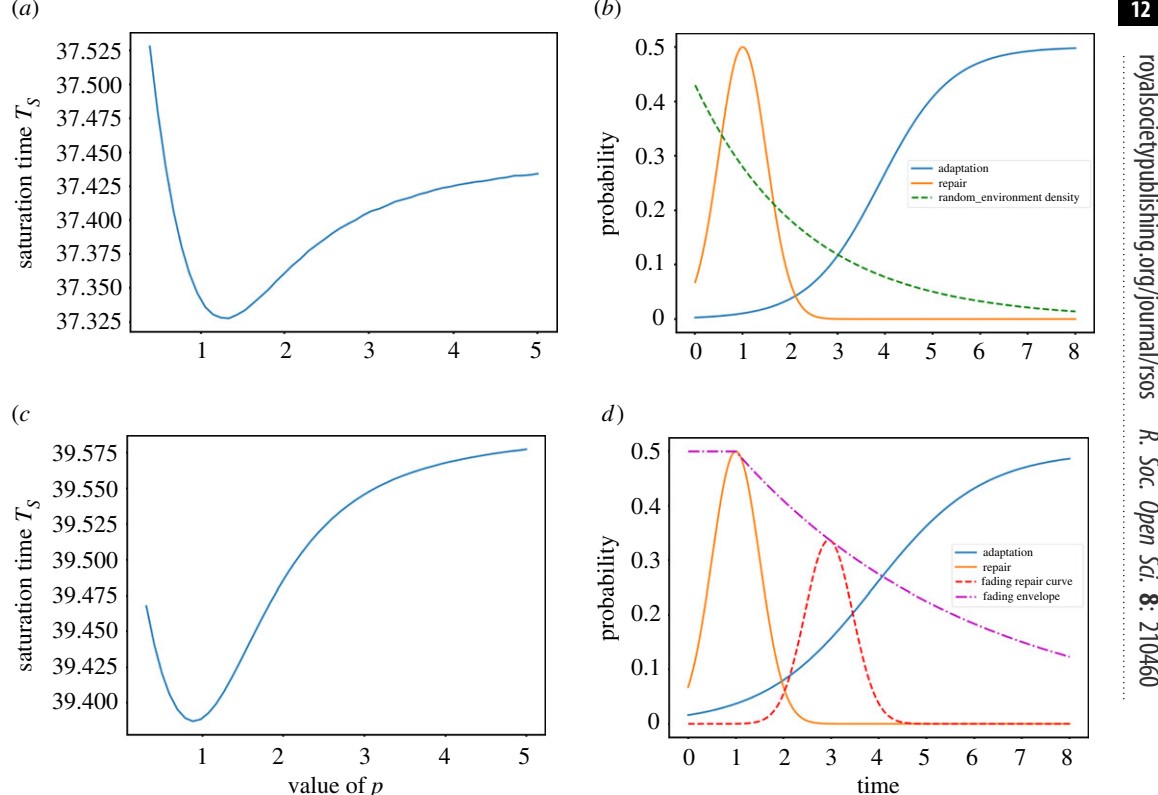

**Figure 9.** (*a*) Saturation time $T_S$ as a function of $p$ in an exponential random environment with parameter 0.43 (see (*b*)), for $\mu_b = 3.886$. (*b*) Adaptation probability for the optimal $p$ from (*a*) (blue) and fixed repair probability (orange). The dashed curve represents the exponential probability density of the random environment. (*c*) Saturation time $T_S$ as a function of $p$ with uniform random delay over [1, 10] and exponential fading of the repair probability (see (*d*)), for $\mu_b = 3.886$. (*d*) Adaptation probability for the optimal $p$ from (*c*) (blue) and initial repair probability (orange). The dashed curve represents one example of delayed repair probability once the source of damage stops and the dash-dot curve represents the exponential fading envelope.

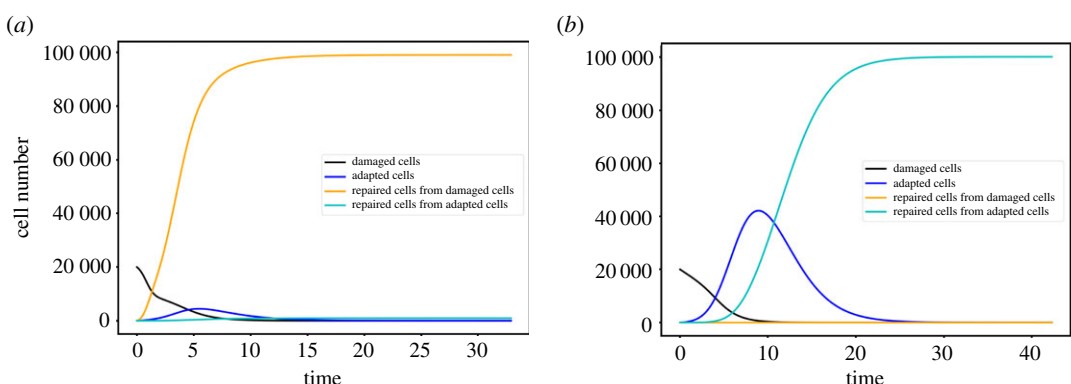

**Figure 10.** (*a*) Evolution in time of the four cell subpopulations (damage cells, adapted cells, repaired cells directly from damaged cells and repaired cells descending from adapted cells), with $\mu_b = 3.886$ and $p = 1$ and no random environment. (*b*) Evolution in time of the four cell subpopulations when direct repair is not allowed but repair of adapted cells is, with $\mu_b = 3.886$ and $p = 1$.

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
