## [Peer Review File · Royal Society Open Science]

Review History

RSOS-210460.R0 (Original submission)

Review form: Reviewer 1

Is the manuscript scientifically sound in its present form?

Yes

Are the interpretations and conclusions justified by the results?

Yes

Is the language acceptable?

Yes

Do you have any ethical concerns with this paper?

No

Have you any concerns about statistical analyses in this paper?

No

Recommendation?

Accept with minor revision (please list in comments)

Comments to the Author(s)

General comments

In this manuscript the authors present the intriguing hypothesis that variance in timing of “adaptation” (i.e. resumption of cellular activity despite lack of DNA repair following damage) may be a bet-hedging strategy that leads to either adaptation or repair under conditions that are temporally unpredictable with respect to efficacy of DNA repair mechanisms. The empirically-informed model is clearly laid out, and provides not only a novel explanation for heterogeneity of cellular adaptation, but also contributes an example arguing for the general importance of considering effects of environmental fluctuations in trait evolution. The manuscript flows well, logic is clearly laid out, and is a pleasure to read.

However, I do have some suggestions for improvement of the manuscript.

1) The continuous model sets out to test the viability of a bet-hedging hypothesis; namely, that adaptation heterogeneity is adaptive under fluctuating conditions. Given that the manuscript centres on a bet-hedging explanation, the idea of bet hedging should be defined, or at least introduced, early on (see Seger, J. & Brockmann, H. J. (1987) What is bet-hedging? *Oxf. Surv. Evol. Biol.* 4, 182-211). Also, predictions for the model outcomes, based on a bet-hedging hypothesis should be presented. Criteria for tests of bet hedging are specifically laid out in Simons, A.M. (2011) Modes of response to environmental change and the elusive empirical evidence for bet hedging. *Proc Roy Soc (B)* 278:1601-1609.

This brings us to a second general point – 2) that the evolution of (heterogeneity of) adaptation should depend on its fitness effects with respect to levels of selection; whether the cell is unicellular or multicellular, or whether germ line or somatic, etc. However, the manuscript is agnostic in this regard, or seems to implicitly treat the cell as the individual/organism and the target of selection. The assumptions being made in this regard should be explicit, especially since this determines what predictions are being made (what is bet hedging for single-celled organisms may in fact simply be optimal cell behaviour for individual multicellular organisms), and some discussion of this issue is merited.

3) The use of the term “adaptation”, to denote resumption of cell cycle progression despite damage, is unfortunate. The awkward conclusion here is that ‘adaptation’ is an adaptation to variable conditions. It may be that historical precedence for this usage is so deep that this word cannot be replaced; otherwise, I suggest this opportunity should be taken to do so.

Minor comments/edits

P. 3, l. 44: “...adaptation timing is important for optimal cell survival...” Cell survival, as a proxy for fitness, is not optimized; it is maximized. Unless this is meant to refer to a balance, e.g. between survival and apoptosis.

P. 4, l. 44-49: Unless I misunderstand, this implies that the probability of repair remains constant regardless of whether repairs were possible in previous steps. Realistically DNA damage proven to be irreparable in adapted cells remains so in future steps. This should be clarified or justified.

P. 4, l. 54-57: “If there is not enough space to double...” To represent relative fitness and for consistency, why not fill the environment to capacity (proportionally) at every step? The potential problem here is that steps with low total population sizes will be underrepresented in

calculations of long-term fitness (i.e. have low weight); as it stands, this is an arbitrary hybrid of absolute and relative fitness.

P. 5 onward: I think the blue/orange colour coding is effective; however, it does require explanation, including in figure captions (e.g. Fig. 2 mentions blue only).

P. 8 (all figures) Axis labels and titles are too small.

P. 14, l. 43 Error/typo in "...has no defect in homology-dependent repair"?

P. 15, l. 21-24 "...rather than strongly favouring heterogeneity, an unpredictable fluctuating environment actually erases the previously found preference for a sharp adaptation timing." I am not convinced that these are not identical statements; i.e. favouring heterogeneity is the same as disfavouring synchronicity. Same comment for subheading 2.4.

P. 15, l.45-48: "Despite a slight..." requires re-wording because the possible contribution of additional factors cannot be said to exist "despite" an original known advantage. Perhaps what is meant is that heterogeneity in adaptation provides only a slight survival advantage, and there may be other selective advantages to heterogeneity.

P. 16, l.10-12, Authors' contributions: The conflicting statements about which author drafted the manuscript should be resolved.

Decision letter (RSOS-210460.R0)

Dear Dr Xu

On behalf of the Editors, we are pleased to inform you that your Manuscript RSOS-210460 "Adaptation to DNA damage as a bet-hedging mechanism in a fluctuating environment" has been accepted for publication in Royal Society Open Science subject to minor revision in accordance with the referees' reports. Please find the referees' comments along with any feedback from the Editors below my signature.

Please submit your revised manuscript and required files (see below) no later than 7 days from today's (ie 23-Jul-2021) date. Note: the ScholarOne system will 'lock' if submission of the revision is attempted 7 or more days after the deadline. If you do not think you will be able to meet this deadline please contact the editorial office immediately.

Please note article processing charges apply to papers accepted for publication in Royal Society Open Science (<https://royalsocietypublishing.org/rsos/charges>). Charges will also apply to papers transferred to the journal from other Royal Society Publishing journals, as well as papers submitted as part of our collaboration with the Royal Society of Chemistry

(<https://royalsocietypublishing.org/rsos/chemistry>). Fee waivers are available but must be requested when you submit your revision (<https://royalsocietypublishing.org/rsos/waivers>).

on behalf of Dr Feng Fu (Associate Editor) and Mark Chaplain (Subject Editor)
openscience@royalsociety.org

Reviewer comments to Author:

Reviewer: 1

Comments to the Author(s)

General comments

In this manuscript the authors present the intriguing hypothesis that variance in timing of “adaptation” (i.e. resumption of cellular activity despite lack of DNA repair following damage) may be a bet-hedging strategy that leads to either adaptation or repair under conditions that are temporally unpredictable with respect to efficacy of DNA repair mechanisms. The empirically-informed model is clearly laid out, and provides not only a novel explanation for heterogeneity of cellular adaptation, but also contributes an example arguing for the general importance of considering effects of environmental fluctuations in trait evolution. The manuscript flows well, logic is clearly laid out, and is a pleasure to read.

However, I do have some suggestions for improvement of the manuscript.

1) The continuous model sets out to test the viability of a bet-hedging hypothesis; namely, that adaptation heterogeneity is adaptive under fluctuating conditions. Given that the manuscript centres on a bet-hedging explanation, the idea of bet hedging should be defined, or at least introduced, early on (see Seger, J. & Brockmann, H. J. (1987) What is bet-hedging? *Oxf. Surv. Evol. Biol.* 4, 182–211). Also, predictions for the model outcomes, based on a bet-hedging hypothesis should be presented. Criteria for tests of bet hedging are specifically laid out in Simons, A.M. (2011) Modes of response to environmental change and the elusive empirical evidence for bet hedging. *Proc Roy Soc (B)* 278:1601-1609.

This brings us to a second general point – 2) that the evolution of (heterogeneity of) adaptation should depend on its fitness effects with respect to levels of selection; whether the cell is unicellular or multicellular, or whether germ line or somatic, etc. However, the manuscript is agnostic in this regard, or seems to implicitly treat the cell as the individual/organism and the target of selection. The assumptions being made in this regard should be explicit, especially since this determines what predictions are being made (what is bet hedging for single-celled organisms may in fact simply be optimal cell behaviour for individual multicellular organisms), and some discussion of this issue is merited.

3) The use of the term “adaptation”, to denote resumption of cell cycle progression despite damage, is unfortunate. The awkward conclusion here is that ‘adaptation’ is an adaptation to variable conditions. It may be that historical precedence for this usage is so deep that this word cannot be replaced; otherwise, I suggest this opportunity should be taken to do so.

Minor comments/edits

P. 3, l. 44: "...adaptation timing is important for optimal cell survival..." Cell survival, as a proxy for fitness, is not optimized; it is maximized. Unless this is meant to refer to a balance, e.g. between survival and apoptosis.

P. 4, l. 44-49: Unless I misunderstand, this implies that the probability of repair remains constant regardless of whether repairs were possible in previous steps. Realistically DNA damage proven to be irreparable in adapted cells remains so in future steps. This should be clarified or justified.

P. 4, l. 54-57: "If there is not enough space to double..." To represent relative fitness and for consistency, why not fill the environment to capacity (proportionally) at every step? The potential problem here is that steps with low total population sizes will be underrepresented in calculations of long-term fitness (i.e. have low weight); as it stands, this is an arbitrary hybrid of absolute and relative fitness.

P. 5 onward: I think the blue/orange colour coding is effective; however, it does require explanation, including in figure captions (e.g. Fig. 2 mentions blue only).

P. 8 (all figures) Axis labels and titles are too small.

P. 14, l. 43 Error/typo in "...has no defect in homology-dependent repair"?

P. 15, l. 21-24 "...rather than strongly favouring heterogeneity, an unpredictable fluctuating environment actually erases the previously found preference for a sharp adaptation timing." I am not convinced that these are not identical statements; i.e. favouring heterogeneity is the same as disfavouring synchronicity. Same comment for subheading 2.4.

P. 15, l.45-48: "Despite a slight..." requires re-wording because the possible contribution of additional factors cannot be said to exist "despite" an original known advantage. Perhaps what is meant is that heterogeneity in adaptation provides only a slight survival advantage, and there may be other selective advantages to heterogeneity.

P. 16, l.10-12, Authors' contributions: The conflicting statements about which author drafted the manuscript should be resolved.

===PREPARING YOUR MANUSCRIPT===

Your revised paper should include the changes requested by the referees and Editors of your manuscript. You should provide two versions of this manuscript and both versions must be provided in an editable format:
 one version identifying all the changes that have been made (for instance, in coloured highlight, in bold text, or tracked changes);
 a 'clean' version of the new manuscript that incorporates the changes made, but does not highlight them. This version will be used for typesetting.

===PREPARING YOUR REVISION IN SCHOLARONE===

-- Ensure that your data access statement meets the requirements at <https://royalsociety.org/journals/authors/author-guidelines/#data>. You should ensure that you cite the dataset in your reference list. If you have deposited data etc in the Dryad repository, please only include the 'For publication' link at this stage. You should remove the 'For review' link.

Author's Response to Decision Letter for (RSOS-210460.R0)

See Appendix A.

Decision letter (RSOS-210460.R1)

Dear Dr Xu,

I am pleased to inform you that your manuscript entitled "Adaptation to DNA damage as a bet-hedging mechanism in a fluctuating environment" is now accepted for publication in Royal Society Open Science.

You can expect to receive a proof of your article in the near future. Please contact the editorial office (openscience@royalsociety.org) and the production office

(openscience_proofs@royalsociety.org) to let us know if you are likely to be away from e-mail contact – if you are going to be away, please nominate a co-author (if available) to manage the proofing process, and ensure they are copied into your email to the journal. Due to rapid publication and an extremely tight schedule, if comments are not received, your paper may experience a delay in publication.

on behalf of Dr Feng Fu (Associate Editor) and Mark Chaplain (Subject Editor)
openscience@royalsociety.org

Appendix A

Dear Editor,

We thank you for handling our manuscript and for your decision. Please find below a point-by-point response (black) to the reviewer's comments (blue).

Sincerely,
Zhou Xu

Reviewer: 1

Comments to the Author(s)

General comments

In this manuscript the authors present the intriguing hypothesis that variance in timing of “adaptation” (i.e. resumption of cellular activity despite lack of DNA repair following damage) may be a bet-hedging strategy that leads to either adaptation or repair under conditions that are temporally unpredictable with respect to efficacy of DNA repair mechanisms. The empirically-informed model is clearly laid out, and provides not only a novel explanation for heterogeneity of cellular adaptation, but also contributes an example arguing for the general importance of considering effects of environmental fluctuations in trait evolution. The manuscript flows well, logic is clearly laid out, and is a pleasure to read.

We thank the reviewer for his/her work and the constructive comments.

However, I do have some suggestions for improvement of the manuscript.

1) The continuous model sets out to test the viability of a bet-hedging hypothesis; namely, that adaptation heterogeneity is adaptive under fluctuating conditions. Given that the manuscript centres on a bet-hedging explanation, the idea of bet hedging should be defined, or at least introduced, early on (see Seger, J. & Brockmann, H. J. (1987) What is bet-hedging? *Oxf. Surv. Evol. Biol.* 4, 182–211). Also, predictions for the model outcomes, based on a bet-hedging hypothesis should be presented. Criteria for tests of bet hedging are specifically laid out in Simons, A.M. (2011) Modes of response to environmental change and the elusive empirical evidence for bet hedging. *Proc Roy Soc (B)* 278:1601-1609.

We have now added a definition of bet-hedging in the introduction. In the discussion, we already provide arguments rejecting adaptation as a sense-and-response strategy (or adaptative tracking). We now recapitulate how adaptation to DNA damage fits some criteria for bet-hedging as defined in the references the reviewer cites and suggest an experimental test of the bet-hedging hypothesis, which would provide a higher level of evidence strength. Incidentally, we have actually found mutations that alter the slope of adaptation probability (manuscript in preparation) and the prediction can thus really be tested experimentally in the future.

This brings us to a second general point—2) that the evolution of (heterogeneity of) adaptation should depend on its fitness effects with respect to levels of selection; whether

the cell is unicellular or multicellular, or whether germ line or somatic, etc. However, the manuscript is agnostic in this regard, or seems to implicitly treat the cell as the individual/organism and the target of selection. The assumptions being made in this regard should be explicit, especially since this determines what predictions are being made (what is bet hedging for single-celled organisms may in fact simply be optimal cell behaviour for individual multicellular organisms), and some discussion of this issue is merited.

We agree with the reviewer that depending on the considered organism subject to selection, the evolution perspective on adaptation to DNA damage is different and might not be applicable. Because adaptation to DNA damage is only relevant at the cellular level, we now explicitly refer to isogenic population of yeast in the introduction. In the discussion, we propose that the conclusions can be extended to other eukaryotic micro-organisms in which adaptation to DNA damage is conserved.

3) The use of the term “adaptation”, to denote resumption of cell cycle progression despite damage, is unfortunate. The awkward conclusion here is that ‘adaptation’ is an adaptation to variable conditions. It may be that historical precedence for this usage is so deep that this word cannot be replaced; otherwise, I suggest this opportunity should be taken to do so.

The term “adaptation” in “adaptation to DNA damage” is indeed unfortunate in the context of this work, which attempts to provide an evolutionary perspective on the matter. However, since it was coined in 1997 by Toczyski et al., it has been widely used by the scientific community working on this topic and has been extended to refer to the override of other types of cellular checkpoints as well (e.g. “adaptation to the spindle assembly checkpoint”). Thus, we cannot replace it with another term. We actually took care not to use “adaptation” in the broad sense of the term, to avoid confusion.

Minor comments/edits

P. 3, l. 44: “...adaptation timing is important for optimal cell survival...” Cell survival, as a proxy for fitness, is not optimized; it is maximized. Unless this is meant to refer to a balance, e.g. between survival and apoptosis.

We have now replaced “optimal” by “maximal”.

P. 4, l. 44-49: Unless I misunderstand, this implies that the probability of repair remains constant regardless of whether repairs were possible in previous steps. Realistically DNA damage proven to be irreparable in adapted cells remains so in future steps. This should be clarified or justified.

Cells that did not repair a damage at a given cell cycle can actually repair it at a later stage. Indeed, it is speculated that alternative repair pathways might be active at other phases of the cell cycle, such as non-homologous end-joining or microhomology-mediated end-joining. Adaptation would give the cell an opportunity to attempt these repairs, despite the fact that homologous recombination in G2 (the most common repair mechanism at this stage) failed. This notion was briefly mentioned in the introduction. But we have now added a sentence in the paragraph describing the stochastic model: “The fact that adapted cells can repair a

damage that was not repaired in the previous cell cycle reflects the idea that alternative repair pathways might be activated at other phases of the cell cycle”.

P. 4, l. 54-57: “If there is not enough space to double...” To represent relative fitness and for consistency, why not fill the environment to capacity (proportionally) at every step? The potential problem here is that steps with low total population sizes will be underrepresented in calculations of long-term fitness (i.e. have low weight); as it stands, this is an arbitrary hybrid of absolute and relative fitness.

We agree with the reviewer and filling the environment proportionally is already what we do. We realize that the misunderstanding comes from the following sentence: “depending on the number n of steps since the damage, damaged cells adapt with a probability $\beta(n)$ and repair their DNA with a probability $\min(\alpha(n), 1 - \gamma_1 - \beta(n))$ – the minimum used here arbitrarily favors adaptation when the sum of $\beta(n)$, $\alpha(n)$ and γ_1 exceeds 1;”

This rule was used initially when we tested probabilities of repair and adaptation that went above 0.5, the sum of which potentially exceeding 1. However, this was no longer the case in the submitted version but we forgot to change the sentence. We apologize for this oversight and the sentence is now corrected, that is, repair probability is directly defined by $\alpha(n)$ and not by $\min(\alpha(n), 1 - \gamma_1 - \beta(n))$.

The code in the github link was already correct.

P. 5 onward: I think the blue/orange colour coding is effective; however, it does require explanation, including in figure captions (e.g. Fig. 2 mentions blue only).

We have now added a sentence in the description of the model explaining the colour code and modified the figure legends where relevant.

P. 8 (all figures) Axis labels and titles are too small.

We have now increased the size of all axis labels and titles.

P. 14, l. 43 Error/typo in “...has no defect in homology-dependent repair”?

Unless we missed something, we do not think there is an error or typo in this sentence. Here, “homology-dependent repair” globally refers to repair mechanisms relying of sequence homology, including homologous recombination, break-induced replication or single-strand annealing. We took the opportunity of this revision to include here another relevant work that we did cite in the first version of the manuscript.

P. 15, l. 21-24 “...rather than strongly favouring heterogeneity, an unpredictable fluctuating environment actually erases the previously found preference for a sharp adaptation timing.” I am not convinced that these are not identical statements; i.e. favouring heterogeneity is the same as disfavouring synchronicity. Same comment for subheading 2.4.

We agree that “favouring heterogeneity is the same as disfavouring synchronicity”. However, here we are already reasoning at the level of heterogeneity, represented by the slope p of the adaptation probability, not at the level of the timing itself. We are saying that

an unpredictable environment no longer leads to an optimal $p > 7$ (sharp adaptation timing), but it does not strongly select for a small value of p either (Fig. 5A, where T_s does not vary a lot numerically). Quantitatively, it seems that cells would fare approximately as well with $p = 1$ as with $p > 4$ (rather sharp adaptation timing). Therefore, we think that an unpredictable environment only slightly favours heterogeneity, but certainly does not select for a sharp adaptation timing ($p > 7$).

P. 15, l.45-48: “Despite a slight...” requires re-wording because the possible contribution of additional factors cannot be said to exist “despite” an original known advantage. Perhaps what is meant is that heterogeneity in adaptation provides only a slight survival advantage, and there may be other selective advantages to heterogeneity.

We agree that the existence of others factors contributing to the selection of heterogeneity is not at odds with the slight survival advantage and we have now replaced “Despite” by “In addition to”.

P. 16, l.10-12, Authors’ contributions: The conflicting statements about which author drafted the manuscript should be resolved.

Pierre Roux and Zhou Xu wrote the first draft of the manuscript together. The statements are now rephrased to reflect this in a single sentence.